# Porous Media Computational Fluid Dynamics and the Role of the First Coil in the Embolization of Ruptured Intracranial Aneurysms

**DOI:** 10.3390/jcm10071348

**Published:** 2021-03-24

**Authors:** Karol Wiśniewski, Bartłomiej Tomasik, Zbigniew Tyfa, Piotr Reorowicz, Ernest J. Bobeff, Ludomir Stefańczyk, Bartłomiej J. Posmyk, Krzysztof Jóźwik, Dariusz J. Jaskólski

**Affiliations:** 1Department of Neurosurgery and Neurooncology, Medical University of Lodz, Barlicki University Hospital, Kopcińskiego 22, 90-153 Lodz, Poland; ernestbobeff@gmail.com (E.J.B.); posmyk.bartlomiej@gmail.com (B.J.P.); dariusz.jaskolski@umed.lodz.pl (D.J.J.); 2Department of Biostatistics and Translational Medicine, Medical University of Lodz, 15 Mazowiecka St., 92-215 Lodz, Poland; bartlomiej.tomasik@umed.lodz.pl or; 3Department of Radiation Oncology, Dana-Farber Cancer Institute, Harvard Medical School, Boston, MA 02215, USA; 4Institute of Turbomachinery, Lodz University of Technology, Medical Apparatus Division, Wolczanska 219/223, 90-924 Lodz, Poland; zbigniew.tyfa@dokt.p.lodz.pl (Z.T.); piotr.reorowicz@p.lodz.pl (P.R.); krzysztof.jozwik@p.lodz.pl (K.J.); 5Department of Radiology—Diagnostic Imaging, Medical University of Lodz, Kopcińskiego 22, 90-153 Lodz, Poland; ludomir.stefanczyk@umed.lodz.pl

**Keywords:** embolization, coils, recanalization, predictors, 1st VPD, computational fluid dynamics

## Abstract

Background: The objective of our project was to identify a late recanalization predictor in ruptured intracranial aneurysms treated with coil embolization. This goal was achieved by means of a statistical analysis followed by a computational fluid dynamics (CFD) with porous media modelling approach. Porous media CFD simulated the hemodynamics within the aneurysmal dome after coiling. Methods: Firstly, a retrospective single center analysis of 66 aneurysmal subarachnoid hemorrhage patients was conducted. The authors assessed morphometric parameters, packing density, first coil volume packing density (1st VPD) and recanalization rate on digital subtraction angiograms (DSA). The effectiveness of initial endovascular treatment was visually determined using the modified Raymond–Roy classification directly after the embolization and in a 6- and 12-month follow-up DSA. In the next step, a comparison between porous media CFD analyses and our statistical results was performed. A geometry used during numerical simulations based on a patient-specific anatomy, where the aneurysm dome was modelled as a separate, porous domain. To evaluate hemodynamic changes, CFD was utilized for a control case (without any porosity) and for a wide range of porosities that resembled 1–30% of VPD. Numerical analyses were performed in Ansys CFX solver. Results: A multivariate analysis showed that 1st VPD affected the late recanalization rate (*p* < 0.001). Its value was significantly greater in all patients without recanalization (*p* < 0.001). Receiver operating characteristic curves governed by the univariate analysis showed that the model for late recanalization prediction based on 1st VPD (AUC 0.94 (95%CI: 0.86–1.00) is the most important predictor of late recanalization (*p* < 0.001). A cut-off point of 10.56% (sensitivity—0.722; specificity—0.979) was confirmed as optimal in a computational fluid dynamics analysis. The CFD results indicate that pressure at the aneurysm wall and residual flow volume (blood volume with mean fluid velocity > 0.01 m/s) within the aneurysmal dome tended to asymptotically decrease when VPD exceeded 10%. Conclusions: High 1st VPD decreases the late recanalization rate in ruptured intracranial aneurysms treated with coil embolization (according to our statistical results > 10.56%). We present an easy intraoperatively calculable predictor which has the potential to be used in clinical practice as a tip to improve clinical outcomes.

## 1. Introduction

Recanalization after endovascular embolization occurs fairly frequently compared with surgical clipping, ranging from 6.1 to 33.6% [1]. Thus, endovascular coiling carries a higher risk of late rebleeding and a lower rate of complete obliteration than surgical clipping [2,3,4,5]. To overcome this shortcoming, in recent years, we have been experiencing breakthrough advances in endovascular treatment of intracranial aneurysms. Dynamic technological progress led to the emergence of new solutions before we could fully appreciate the safety and effectiveness of those already accepted [6,7,8]. This fact hinders homogeneous assessment of the selected endovascular treatments. 

There are still no clear criteria as to which type of endovascular procedure should be employed in which case. The choices relating to coil characteristics, the use of a stent, flow diverter or flow disrupters depend primarily on the judgment of the medical team. The main objective of the present study was to draw attention to the problem of recanalization in ruptured aneurysm. We wanted to emphasise the need for the appropriate patient qualification to a specific endovascular procedure in order to reduce the risk of recanalization. According to the Cerebral Aneurysm Rerupture After Treatment (CARAT) study, late retreatment was more common after coil embolization than after clipping. Furthermore, the degree of aneurysm occlusion after the initial treatment was a strong predictor of the risk of subsequent rupture in patients presenting with SAH. The CARAT study showed that endovascular coiling presents a high incidence of late rebleeding and a low rate of complete obliteration [9,10]. 

To date, many factors associated with the recanalization of aneurysms treated with coil embolization have been presented, namely aneurysm morphological features, the Raymond–Roy scale score at immediate angiography, aneurysm location, packing density (PD), coil compaction, and different types of endovascular embolization techniques [11,12,13,14,15,16,17]. Some studies using computational fluid dynamics (CFD) also suggested a key role of cerebral hemodynamics in recanalization [18,19,20]. Many studies concluded that it was necessary to increase the PD by at least 20–24% to avoid aneurysmal recanalization [11,12,21]; however, the final decision on coil characteristics and number of coils used during embolization always depends on the neuroradiologist. On the other hand, some authors claimed that the first coil is crucial for obtaining a high PD and improving outcomes [22,23]. It is still unknown how the first coil changes intraaneurysmal hemodynamic parameters and how it influences recanalization. The mechanistic background of the first coil volume packing density (1st VPD) was not well investigated. 

In this study, we looked at the role of 1st VPD in the embolization of ruptured intracranial aneurysms (RIAs) in order to identify a strong predictor for recanalization after coil embolization of ruptured aneurysms and to find the basic mechanistic/physical background of our clinical observations. This goal was achieved by means of a statistical analysis followed by a computational fluid dynamics (CFD) with a porous media modelling approach, during both stationary and transient simulations (mimicking pulsatile blood flow).

## 2. Material and Methods

### 2.1. Ethics Approval

The study was approved by the local Bioethical Committee at the Medical University of Lodz, application number RNN/119/15/KE. All experiments were performed in accordance with relevant guidelines and regulations. The study was designed in accordance with the Good Clinical Practice (GCP) guidelines and was conducted according to the principles of the Declaration of Helsinki. Informed consent was obtained from the participants prior to inclusion. In case of depressed level of consciousness, the patient’s legal representative was asked for informed consent.

### 2.2. Patients

The authors made a retrospective analysis of digital subtraction angiograms (DSA) and reviewed the notes of 102 patients with burst intracranial aneurysms treated with coil embolization over a 2-year period. In the next step, we simulated our statistical results using computational fluid dynamics (CFD) with porous media modelling approach. Porous media CFD simulated the hemodynamics within aneurysmal dome after coiling.

All the patients had an angiography immediately followed by embolization and then had 6- and 12-month check-up DSA. 

The decision on the method of treatment was made by neurosurgeons, neuroradiologists and anesthetists. In case of any doubts, embolization was the treatment of choice.

On admission to the hospital, the patients were assigned according to the Hunt and Hess scale (HH scale). Demographic and medical data (age, gender, comorbidities and laboratory results) were entered into medical records. SAH was assessed according to the modified Fisher’s scale. The outcomes were assessed using the GOS scale at discharge and with modified ranking scale after 12 months. Details are presented in Table 1.

### 2.3. Digital Subtraction Angiography/Embolization

A written consent for the procedure was obtained from all conscious patients. If they were unable to sign the consent, the patients gave their consent orally in the presence of two witnesses, which was noted in medical records. In the case of unconscious patients with high risk of death, consent was not obtained and a protocol of necessity was drawn up.

Endovascular procedures were performed by one of two experienced interventional radiologists using a Siemens Axium Artis apparatus (Siemens, Appleton, WI, USA).

The procedure was performed via the femoral artery. For angiography, the 5F introducer was inserted, through which a hydrophilic guide (0.035 inch) and a catheter were passed. Both internal carotid and both vertebral arteries were catheterized for selective angiography; non-iodine contrast agent was administered from an automatic syringe—10 mL at 5 mL/s, into carotid arteries and 7 mL at 4 mL/s into vertebral arteries. After performing angiography in four projections, a working projection, i.e., the one that best showed the aneurysm’s morphology, was always selected. A calibration of the catheter located in the carotid or vertebral artery was done on the basis of digital images of subtraction angiography obtained in the anteroposterior and lateral projections with unified enlargement up to 22 cm. After this, the measurements of the aneurysm were taken, the ratio of its sac to the neck calculated and anatomical relations to the adjacent vessels were assessed 

The endovascular treatment was performed under general anesthesia. Patients received 5000 units of unfractionated heparin intravenously (Pfizer, New York, NY, USA), followed by 2000 units for each hour of the procedure. The distal end of the Chaperon guide catheter (MicroVention, Aliso Viejo, CA, USA) (5F or 6F, depending on the diameter of the vessel) was placed in the first (carotid-C1) segment of the internal carotid artery (in the case of the vertebral artery-in segment V2). The guide catheter was flushed with 0.9% saline solution with heparin at 2000 units/L. The procedure was performed using the so-called roadmapping technique—after administration of the contrast agent, a Marathon 10 microcatheter (Medtronic, Minneapolis, USA) with a Traxcess 14 microguide (MicroVention, Aliso Viejo, CA, USA) was inserted into the aneurysmal sac. Platinum embolization spirals, hydraulically or mechanically released, or electrolytically detachable spirals coated with hydrogel were introduced successively. Their diameter was 0.010″ and 0.018″, and they were available in two forms of coil: spatial (3D) or helical [24].

Assessment of Aneurysm Morphometric Parameters, Packing Density, 1st VPD, Recanalization and Degree of Aneurysm Filling during the First Embolization.

The morphometric parameters which were taken into consideration were: aneurysm dome height (H), width (S), and depth size (D);aneurysm neck size (N);parent artery size (P);the largest aneurysm dimension perpendicular to the neck (H_max_);aspect ratio (AR), defined as the maximal perpendicular height (the largest perpendicular distance from the neck of the aneurysm to the dome of the aneurysm) divided by neck width;size ratio (SR), defined as maximum aneurysm height (between the center of the aneurysm neck and the greatest distance to the aneurysm dome), divided by the parent artery diameter;the index determining the ratio of aneurysm neck width to diameter of the parent artery.

The diameter of the aneurysm was determined on the basis of its largest diameter, expressed in millimeters (Figure 1).

The presence of recanalization was determined on the base of DSA digital images. The volume of the aneurysm was calculated using the following formulas tailored to its shape (spherical, ellipsoid or bilobed):

Spherical shape:(1)$$volume=\;\frac{\pi \cdot diameter^{3}}{6}$$

Ellipsoid shape:(2)$$volume=\;\frac{\pi \cdot diameter^{2}\cdot height}{6}$$

Bilobed shape: ellipsoid A + ellipsoid B 

The spiral volume was calculated according to the formula:(3)$$volume=\;\frac{\pi \cdot diameter^{2}\cdot length}{4}$$

The packing density was calculated according to the formula:(4)$$packing\;density=\;\frac{spiral\;volume}{aneurysm\;volume}\cdot 100\%$$

Two spiral diameters were taken into account: 0.″10″ and 0.″18″. The volume of the aneurysm was calculated in cubic millimeters, whereas the packing density and the 1st VPD were calculated as percentages.

The effectiveness of endovascular treatment was assessed visually after embolization and during follow-up DSA after 6 and 12 months. The modified Raymond–Roy scale was used to assess recanalization. Recanalization was noted if there was an increased aneurysm filling compared with the examination performed immediately after embolization [24].

### 2.4. Statistical Analysis

Nominal variables were analyzed using the chi-square test. Continuous variables were presented as the mean with standard deviation or median with interquartile range (25–75 percentile), depending on the data distribution, which was verified using the W Shapiro–Wilk test. Intergroup differences for variables with normal distribution were analyzed using the t-test, and for variables with non-normal distribution, differences were measured using the non-parametric Mann–Whitney U test. Correlations were assessed using the Spearman’s rank correlation test. Receiver operating characteristic (ROC) curves were utilized to evaluate the quality of the classifiers and identify the optimal cut-off point. In the multifactor analysis, the stepwise logistic regression was used. The statistical analysis was performed using the Statistica 13.3 package (TIBCO Software Inc., CA, USA) and *p* < 0.05 was considered significant.

### 2.5. Computational Fluid Dynamics Analysis

To support the statistical analysis presented in this research, the authors performed thorough CFD investigations based on a single patient-specific geometry. A case study for the numerical analyses was selected among all patients and included an aneurysm located at the basilar artery bifurcation. The CFD study was performed with a porous media modelling approach, i.e., the aneurysm dome was modelled as a separate porous domain, where specific parameters mimicking varied first coil packing density (1st VPD) were iteratively analyzed. 

The initial patient-specific geometry was prepared in a custom-created software (*Anatomical Model Reconstructor, AMR*) basing on biomedical images acquired during standard angio-CT examination. This software is continually being developed at the Institute of Turbomachinery at Lodz University of Technology (Poland), and thus, currently it is only an in-house program.

Despite the possibility of reconstructing the entire cerebral vasculature, it was decided to limit the geometry only to the basilar artery aneurysm together with efferent and afferent vessels (vertebral, basilar, posterior cerebral, and posterior cerebellar arteries). Thus, a direct influence of the varied packing density parameter could be observed. 

The digital patient-specific surface model was obtained from a 3D binary mask that was created during an automatic image segmentation technique known as a region growing (or seed growing). Voxels representing blood domain, which were intensity-matched with predefined thresholding criteria, were iteratively added to an initial region selected by the user. To enhance the output of such an automatic image segmentation method, manual 3D mask processing was required. Fortunately, AMR software offers an intuitive approach towards manual manipulation of the binary mask—the user can add or remove a region selected with either a lasso or a brush. Thus, if any unnecessary anatomical structure (such as bone fragment or another blood vessel that was too close to the investigated artery) was added to the mask during the seed growing method, the authors could easily select and remove it. Afterwards, a surface model could be generated from the spatial mask thanks to the marching cubes algorithm. Due to the fact that this surface was a voxelated one (characterized by a stair-like structure), it was subjected to an automatic smoothing algorithm, available in AMR software as well. This strives for smoothing the surface wall while minimizing topology changes of the entire structure. Finally, distal parts of the efferent and proximal parts of the afferent vessels were clipped to obtain an opened model whose inlet and outlet cross sections were as perpendicular to the flow channel as possible. This was performed in AMR software as well. The generated surface model was stored in stereolithography (STL) format, which is extremely difficult to manipulate—for instance, a proper separation of the aneurysm dome from the parent vessel is a troublesome and laborious task. Thus, the authors used ANSYS SpaceClaim module to convert the surface model into a volumetric one. Afterwards, this geometry was imported into SolidWorks software, where the aneurysm dome was separated from the parent vessel.

In all the numerical simulations presented within this research, blood was modelled as an incompressible fluid of density equal to 1050 kg/m^3^. Additionally, its non-Newtonian shear-thinning behavior was described by a modified power law viscosity model, governed by the following formula:(5)$$\left\{\begin{array}{c} \begin{array}{ll} \eta =0.55471\;Pa\cdot s & \mathrm{for}\;\dot{\gamma }\leq 0.001\;\\ \eta =\;\eta_{0}\cdot {\left(\dot{\gamma }\right)}^{n-1} & \mathrm{for}\;0.001\;\leq \;\dot{\gamma }<327\\ \;\eta =0.00345\;Pa\cdot s & \mathrm{for}\;\dot{\gamma }\geq 327 \end{array} \end{array}\right.$$
where $\eta_{0}$ = 0.035 [kg∙m^−1^∙s^−1.4^]; n = 0.6 (-)

The flow was assumed as adiabatic and isothermal one (no heat exchange). To solve the Navier–Stokes equations governing the steady-state fluid motion, a commercial solver ANSYS CFX 20.1 (Ansys, Cannonsburg, MI, USA) was used. Navier–Stokes equations are always supplemented with the conservation of mass theorem which is described by the following formula:(6)$$\frac{d\rho }{dt}+\nabla \cdot \left(\rho \vec{U}\right)=0$$
where ρ is the fluid density and $\vec{U}$ is the velocity vector.

Bearing in mind the incompressibility of the analyzed fluid flow, the general vector form of the solved Navier–Stokes equations can be simplified to the following form:(7)$$\frac{d\vec{U}}{dt}=\;{\vec{F}}_{m}-\frac{\nabla p}{\rho }+\;\frac{\mu }{\rho }\nabla^{2}\vec{U}$$
where ${\vec{F}}_{m}$ represents the external forces acting on the fluid, while ∇p is the pressure gradient and μ is the dynamic viscosity.

For the aneurysm dome that was modelled as a porous domain, the flow was additionally governed by Darcy’s Law, i.e., pressure was balanced by resistance forces. Such relation and all governing formulas are presented below:(8)$$\;\nabla p=K\cdot \vec{U}\;\;K=\;\alpha \cdot \left|\vec{U}\right|+\;\beta \;\alpha =\frac{1.75\cdot \rho \cdot \left(1-\kappa \right)}{\kappa^{3}\cdot D_{p}}\;\beta =\frac{150\cdot \mu \cdot {\left(1-\kappa \right)}^{2}}{\kappa^{3}\cdot D_{p}{}^{2}}\;\kappa =1-\frac{CPD}{100}\;CPD=\;\frac{coil\;volume}{aneurysm\;dome\;volume}\;\cdot 100\%$$
where K is a porous resistance constant, α is a quadratic resistance coefficient, β is a linear resistance coefficient, κ is the aneurysm dome’s porosity, $D_{p}$ is the diameter of a coil wire (2.54 × 10^−4^ m), and μ is a critical viscosity value (0.00345 Pa∙s), while CPD is the coil packing density.

To follow the gold standard related to CFD investigations, a mesh independence test was conducted prior to performing target numerical analyses. Its main objective was to choose the most optimal meshing parameters and to ensure that the numerical results do not change significantly with the increasing mesh density. Each mesh comprised tetrahedral elements and an inflation layer composed of prism elements. Table 2 outlines the results of the mesh independence test.

As can be seen, the results differences between the mesh of middle density and the finest mesh are negligible. Thus, the middle-density mesh comprising 6.24 million elements was chosen for the further simulations. Figure 2 depicts the analyzed geometry with marked boundary conditions (for stationary analyses) and indicates fluid domains together with time-dependent inflow boundary condition used for the transient simulations. However, time-dependent inflow velocity was limited only to a single cardiac cycle (for a higher clarity)—in transient simulations, 5 full cycles were simulated. It is worth mentioning that the model wall was assumed to be a rigid one.

To quantitatively evaluate influence of the coil volume packing density (VPD) on the blood inflow to the aneurysm dome, the authors performed numerous simulations, i.e., one control case (without any porosity) and wide range of porosities that resembled 1–30% of VPD, with 1% increment. Such an approach was used for both stationary and time-dependent transient simulations.

## 3. Results

### 3.1. Patients

A total of 102 patients were screened for eligibility and 66 were included. The study flow-chart is presented in Figure 3.

Patient assessment is presented in Table 3.

### 3.2. Aneurysm Location

Among 66 ruptured aneurysms, 42 (63.6%) were located in the anterior part of the circle of Willis. There were 11 aneurysms on ACoA, 4 on MCA, 27 on ICA, including 12 PCoA aneurysms (no fetal type PCoA), 8 on ICA bifurcation, 4 on the ophthalmic segment, and 3 on the anterior choroid artery. 

Twenty-four aneurysms (39.4%) were located in the posterior part of the circle of Willis with 18 of them on BA (bifurcation), 1 aneurysms on PICA, 1 on the posterior cerebral artery, 1 on the superior cerebellar artery, and 3 aneurysms located on the vertebral artery.

Basing on our research and statistical database, we did not find any correlation between aneurysm location and recanalization.

### 3.3. Laboratory Results, Morphometric Parameters, Packing Density, 1st VPD of Intracranial Aneurysms

In the univariate analyses comparing patients with and without recanalization, we observed significant differences between: aneurysm height (9.49 ± 5.19 mm vs. 6.86 ± 3.22 mm, *p* = 0.016), aneurysm neck size (4.14 ± 0.66 mm vs. 3.27 ± 0.83 mm, *p* < 0.001), aneurysm volume (349.15 ± 432.39 mm^3^ vs. 166.48 ± 274.3 mm^3^, *p* = 0.045), packing density (21.2 ± 6.6% vs. 35.0 ± 10.8%, *p* < 0.001), 1st VPD (10.51 ± 2.83% vs. 18.28 ± 4.16%, *p* < 0.001), the largest aneurysm size (9.44 ± 5.17 mm vs. 7.04 ± 3.16 mm, *p* = 0.026, index determining the ratio of neck width to diameter of the parent artery (1.16 ± 0.46 vs. 0.92 ± 0.29, *p* = 0.015), Hmax (12.03 ± 5.09 mm vs. 6.89 ± 3.41 mm, *p* < 0.001) and aspect ratio (3.03 ± 1.59 vs. 2.14 ± 0.90, *p* = 0.006). After Bonferroni correction for multiple hypothesis testing, the differences for neck size, packing density, 1st VPD, and H_max_ remained statistically significant. Detailed results of all performed analyses are presented in Table 4.

### 3.4. Complete Aneurysm Filling during the First Embolization

Complete aneurysm filling on the first embolization according to the modified Raymond–Roy scale took place in 52 cases (class I—78.8%). Incomplete aneurysm filling was noted in 14 cases; class II—11 cases (16.7%), class IIIa—1 case (1.5%) and class IIIb—2 cases (3%).

### 3.5. Type of Embolization Material

Platinum spirals were used in 50 cases, whereas bioactive spirals were used only in 16 cases.

### 3.6. Recanalization

#### 3.6.1. Early Recanalization (after 6 Months)

Recanalization was found in 18 cases (27.3%); in 12 cases in grade 3b according to the Raymond–Roy scale and in 6 cases in grade 3a. The treatment was repeated in 18 patients.

#### 3.6.2. Late Recanalization (after 12 Months)

Out of 66 patients, late recanalization was noted in 11 cases (16.6%). In this group, we observed five new recanalized aneurysms (which were not seen in the first control DSA after 6 months). Altogether, there were seven cases in grade 2, according to the Raymond–Roy scale, three cases in grade 2a, and one in grade 2b.

### 3.7. Statistical Analysis

#### 3.7.1. Late Recanalization

The results of the univariate analysis are presented in Table 4.

#### 3.7.2. Morphometric Parameters

Details of multivariate analysis are described in Table 5

The percentage of 1st VPD turned out to be the most important predictor of late recanalization. Its value was significantly greater in all patients without recanalization (Table 4).

Next, we dichotomized the functional outcome scales based on the dependence on permanent help (GOS scores 4–5 vs. 1–3 and mRS scores 0–2 vs. 3–6). We analyzed the correlation between 1st VPD values and clinical markers (e.g., age, sex, aneurysm location, Hunt and Hess, Fisher, GOS, mRS) and we did not find any correlation. 

Subsequently, we verified the predictive value of 1st VPD marker with ROC curves (AUC 0.94 (95%CI: 0.86–1.00) (Figure 4). A cut-off point of 10.56% presented satisfactory discriminatory ability (sensitivity—0.722; specificity—0.979) and was validated in a computational fluid dynamics analysis.

In the univariate analysis, we investigated which combination of the morphometric parameters offered the highest prognostic value regarding the occurrence of late recanalization. Receiver operating characteristic curves based on univariate analysis showed that aneurysm neck size as a single factor represented the best model for late recanalization prediction (AUC 0.83 (95%CI: 0.72–0.95) (Figure 5).

In the next step, we verified the predictive value of packing density marker with ROC curves (AUC 0.88 (95%CI: 0.79–0.98) (Figure 6).

Afterwards, we built a model for 1st VPD to check its predictive value and we received even better results, showing a substantial improvement in the predictive value for the late recanalization prediction (AUC 0.94, 95%CI:0.86–1.00, *p* < 0.001. (Figure 7).

Receiver operating characteristic curves based on backward stepwise logistic regression showed that the best model for late recanalization prediction based on morphometric parameters contained only neck size (AUC 0.83 (95%CI: 0.72–0.95). When we added packing density to this model, we received better results—AUC = 0.88 (95%CI: 0.79–0.98). This improves the outcomes; however, when we added to this model 1st VPD, we received even better results—AUC 0.94 (95%CI:0.86–1.00). Hence, paying attention to 1st VPD in the future, we could improve outcomes (*p* < 0.001).

In a further step, we performed multivariate analysis which showed that only 1st VPD value is statistically significant for the late recanalization (Table 5).

### 3.8. Computational Fluid Dynamics Analysis

The authors prepared the patient-specific model of the aneurysm at the basilar artery bifurcation together with vertebral, basilar, posterior cerebral, and posterior cerebellar arteries. In total, 31 steady-state and 7 transient numerical simulations were conducted, where different porosities (resembling 0–30% of volume packing density, VPD) were assumed.

The most important hemodynamic parameters related to the possible aneurysm rupture were analyzed. They included WSS, TAWSS, OSI, and pressure distribution at the artery and aneurysm walls. Moreover, the authors investigated the ratio of residual flow volume (RFV) and aneurysm volume (AV). RFV is known as a parameter that defines the volume of the fluid domain within the aneurysm where blood velocity exceeds 0.01 m/s [25]. The ratio between RFV and AV indicates how much volume of ‘active blood’ circulates within the aneurysm dome – the lower this value is, the lower the blood velocity is and the larger the possible stagnation zones are. As a result, there are higher chances of possible thrombogenic process occurrence, which can strengthen the wall and lead to the successful aneurysm treatment. The numerical results of the aforementioned parameters are outlined in Table 6 as well as in Figure 8, Figure 9, Figure 10, Figure 11 and Figure 12.

As can be observed, WSS dependence on VPD is characterized by an asymptotic function. This means that after exceeding a certain threshold value, a further increase in VPD does not have a significant influence on the obtained WSS results. This can be seen by analyzing WSS values corresponding to >20% VPD—there is a very flat slope of the decrease in WSS values. The steepest slope can be noticed for the initial values of VPD, i.e., below 5%. It means that the highest differences in WSS magnitude are obtained for lower VPD values.

RFV/AV dependence on VPD is characterized by a sigmoid function (double-side asymptotic function). It means that RFV/AV ratio for the marginal values of VPD is relatively constant, whereas the highest changes occur in the range 10–20% of VPD. Within these threshold values, one can observe the highest inflow reduction to the aneurysmal dome. Thus, it suggests that the 1st VPD value should be in the 10–20% range, which is in agreement with our statistical results (>10.56%).

Dependence of area-averaged velocity on the aneurysm neck and pressure at the aneurysm wall exhibited different characteristics (see Figure 9). Contrary to the former analyses, pressure at the aneurysm wall is characterized by a more complex function with a visible global minimum, global maximum, and a flat region of constant values. The lowest area-averaged pressure at the aneurysmal wall was obtained for 2% of VPD. After this value, there is a rapid increase in pressure as VPD increases up to 10%. Then, pressure starts to decrease asymptotically until it reaches a constant value of circa 11,345 Pa. Some similarities can be observed for area-averaged velocity at the aneurysm neck—initially, a relatively steep reduction in velocity occurs, up to circa 3% of 1st VPD. Then, inflow velocity continually increases (with a slight increment) up to 10% of 1st VPD. After exceeding the 10% threshold, area-averaged velocity starts to gradually decrease with an increase in VPD.

An assumed hypothesis concerning such pressure characteristics is as follows. Despite an initial increase in VPD (2–10% range) and reduction in the blood velocity within the aneurysm dome, inflow to the aneurysm remains nearly constant—the same blood volume gets to the aneurysm; however, due to the artificial coiling, it has a smaller space to propagate through. Hence, the pressure at the aneurysmal wall increases. Almost constant inflow could be proved by investigating the area-averaged velocity at the aneurysm neck—hardly any differences could be noted for successive values of 1st VPD. The local maximum of area-averaged velocity (at 10% of VPD) matched the local maximum of pressure at the aneurysm wall. After exceeding the 10% VPD threshold, the inflow to the aneurysm dome begins to be inhibited, leading to a continuous decrease in the pressure. When reaching the second threshold VPD value, circa 27%, pressure becomes constant. This means that inflow to the aneurysm dome is not changing with a further increase in the coiling volume. It was decided not to perform additional analyses of 1st VPD exceeding 30%, since it was impossible to fill such a volume using a single coil.

Global maximum pressure, 11,356.1 Pa, obtained at 10% of VPD, might suggest that at these settings, the aneurysm has the greatest chances to rupture. However, as can be seen from Table 6 and Figure 9, numerical differences in pressure among successive and marginal VPD values are very low—the entire measured range is equal to circa 14 Pa (11,342–11,356 Pa). 

Figure 10, Figure 11, Figure 12 and Figure 13 depict graphical representations of hemodynamic parameters distribution (including velocity at the aneurysm neck) at given regions computed during stationary simulations.

As can be seen, the distribution of each analyzed hemodynamic parameter continually changes with an increase in 1st VPD. By qualitatively investigating velocity distribution within the aneurysm dome (Figure 12) and at the aneurysm neck (Figure 13), one can clearly see that blood inflow to the aneurysm is inhibited—the higher the 1st VPD values, the larger the inhibition. Reduced inflow to the aneurysm dome results in changes in other hemodynamic parameters, including WSS and pressure characteristics.

As far as transient simulations are concerned, they were limited to the following VPD values: 0%, 5%, 10%, 15%, 20%, 25%, and 30%. It was decided to present data related to area-averaged and maximum WSS values at systole peak (timestep equal to 3.376 s) together with TAWSS and OSI calculated for the last full cardiac cycle. For greater clarity, these data are outlined as plots (see Figure 14).

Focusing on area-averaged WSS distribution at the aneurysm wall (during systole peak) and area-averaged TAWSS values, one could notice similar characteristics as for the stationary simulations. Shear stresses tend to decrease asymptotically towards constant values, which means that any further increase in VPD might not result in significant changes to the WSS or TAWSS parameters. When the OSI parameter is analyzed, one can observe a different characteristic. Initially, it increases up to 0.264 at 10% of 1st VPD and then begins to gradually decrease. This means that by reaching a specific threshold value of 1st VPD (in terms of the analyzed patient-specific geometry, it was 10%), the OSI parameter would be the highest. The authors assume that high OSI is desirable for the aneurysm treatment. In general, the OSI parameter describes fluctuations of the flow—it varies from 0, when the direction of the WSS vector is the same as the direction of the flow, to 0.5, when both directions are opposite. It is claimed that high local OSI values, together with low WSS, show regions prone to plaque formation and instances of blood stagnation [26,27]. Wall portions prone to elevated OSI values might indicate regions where blood could clot. The results of numerous transient simulations indicate that the most optimal conditions for the aneurysm treatment were obtained for 10% of 1st VPD, which is in agreement with other numerical results and proves the correctness of our statistical analysis.

Moreover, to compare a time-dependence of the RFV/AV ratio within the aneurysm dome throughout the entire cardiac cycle, a sequence of plots was prepared (see Figure 15). Cycle-averaged values of RFV/AV ratio are outlined as well. 

As can be seen, the RFV/AV ratio and its time-dependent fluctuations are strictly related to VPD and cardiac cycle phase. The higher the 1st VPD value, the higher the RFV/AV’s proneness to fluctuations. When there is no artificial coiling within the aneurysm dome, the RFV/AV ratio is almost constant and equal to 100%. This means that blood was characterized by velocity exceeding 0.01 m/s within the entire aneurysm dome, notwithstanding the cardiac cycle phase. When the VPD value was increased up to 10%, one could notice a slight pulse of RFV/AV ratio that resembled a slightly dampened pulsatile waveform. With the further increase in 1st VPD value, this pulsatile characteristic began to be more noticeable. Furthermore, it cannot be ignored that when cycle-averaged values are taken into account, one can observe similar characteristics as for the stationary simulations—RFV/AV dependence on 1st VPD exhibits a sigmoid characteristic. Upon exceeding 10% of 1st VPD, the RFV/AV ratio begins to decrease with the steepest slope. After reaching 20–25% of VPD, its slope starts to become more flattened. This is in conformity with the results obtained for stationary simulations. 

Figure 16, Figure 17, Figure 18, Figure 19 and Figure 20 depict graphical representations of hemodynamic parameters’ distribution at aneurysm walls computed during transient simulations.

## 4. Discussion

This is the first study, based on retrospective analysis and porous media CFD, which demonstrates that 1st VPD is a strong and potentially clinically useful recanalization predictor, and it determines the volume of postcoiling intra-aneurysmal residual blood flow.

Late recanalization promotes rebleeding, which is the main cause of death, and thus in this study, we focused on the late recanalization issue. 

The risk of rerupture after RIA coiling was evaluated in several studies. 

The CARAT trial showed that the risk of rebleeding after aneurysm coiling is significantly associated with the quality of aneurysm occlusion. The risk of rebleeding is 1.1% in the case of complete occlusion, 2.9% if occlusion is between 91 and 99%, 5.9% when aneurysm occlusion is between 70 and 90%, and 17.6% when it is less than 70% [9,10].

Instances of rebleeding after the first endovascular treatment were also noted in the ISAT, study and their likelihood was 3.2% [2].

According to ARETA study, the risk of rebleeding 1 year after coiling was estimated at 1.0%. Aneurysm occlusion and dome-to-neck ratio were the two factors that had appeared to play a role in the rebleeding [28].

At present, the literature divides recanalization predictors into three factors: patient-related, aneurysm, and treatment-related.

The CLARITY trial pointed out that the evolution of aneurysm occlusion was significantly affected by age of the patients, but the investigators did not find a correlation between gender and recanalization [29]. One study showed that smoking was another recanalization risk factor [30].

Several studies suggested that ruptured aneurysms were more prone to recanalization than unruptured ones [15,31,32]. Some authors underscored that aneurysms of the basilar tip and those larger than 10mm tend to recanalize [1,15,33]. In addition, some morphometric parameters seem to promote recanalization, particularly the wide-neck, should it reach or exceed 4 mm in size [1,15,29,31,32].

An incomplete occlusion on the first procedure is also a strong predictor of recanalization, as confirmed by most of the studies on the subject [15,31,34,35]. Some authors also drew attention to the importance of packing density, pointing out that a value less than 24% promotes recanalization [36]. 

As recanalization is an important shortcoming of coil embolization treatment, some innovative techniques in endovascular treatment (surface-modified coils, balloon-assisted coiling, stent-assisted coiling, flow diverters (FDs), and flow disrupters) were developed to overcome this problem [37,38,39,40,41,42,43,44]. However, it seems that the modified coils have no influence on recanalization [37,45]. There are no studies evaluating a role of balloon-assisted coiling for recanalization, and there are contradicting data on the stent-assisted coiling and recanalization rate [40,46,47]. Still little is known about recanalization after implantation of FDs or flow disrupters. An important drawback of the above-mentioned techniques is the necessity of the following dual antiplatelet therapy which hampers thrombosis of the aneurysm. 

There are no specific guidelines as to which endovascular procedure should be used in a particular case. In an attempt to address this issue, in this study we looked at the role of 1st VPD in efficacy of coil embolization of ruptured aneurysms, and to visualize our statistical results using porous media CFD.

We examined a number of morphometric parameters, which turned out to be significant in univariate analysis, but out of them, only 1st VPD proved to be significant in multivariate analysis. We looked at packing density and 1st VPD showing their significance in multivariate analysis (Table 5). The ROC curves, plotted for the prediction of recanalization and 1st VPD levels, had AUC > 0.94, indicating the potential clinical value of this marker.

Some studies already reported that the selection of the first coil is an important factor for reaching a high VPD [23,24,48]. First VPD was first introduced by Neki et al. as a predictor of recanalization. The authors considered that the values of 1st VPD which ought to be achieved in order to avoid recanalization should remain between 17.5% and 20% [49].

Thus, in the next step, we performed a porous media CFD analysis to find changes in hemodynamics. In the literature, this method was used to simulate the hemodynamics after aneurysm coiling [50,51]. Our analysis showed that, after exceeding 1st VPD of 10%, pressure at the aneurysm wall and RFV/AV ratio are continually reduced (in both stationary and transient simulations), while the intra-aneurysm hemodynamic postcoiling parameters stabilize. The results are quite similar if we compare them to our retrospective analysis; upon reaching 1st VPD greater than 10.56%, the recanalization rate decreases.

According to the literature, RFV should be kept below 20.4 mm^3^ to prevent recanalization [25]. In our opinion, appropriate 1st VPD decreases the volume of post-coiling intra-aneurysmal residual blood flow, thus preventing recanalization. 

It was also mentioned that partially embolized aneurysms with a high WSS and high blood flow velocity likely recurred during the follow-up [18]. This means that an insufficient flow reduction promotes post-coiling recurrence. Our CFD analysis confirmed that the highest differences in WSS were obtained for lower VPD values, as had been previously noted by Luo et al. [18]. Additionally, the results related to the OSI parameter, which was the highest for 10% of 1st VPD, indicate that this coil packing density promotes the optimal conditions for the aneurysm treatment.

We also observed that 1st VPD levels were quite similar in ruptured aneurysms located both in anterior and posterior circulation (mean—16.23% vs. 16.05%), which probably means that it is easy to achieve similar treatment results for both those aneurysm locations.

The risk of late recanalization may be more effectively predicted than the long-term outcome. Although it is well known that the late outcome depends on recanalization, we did not observe it in our analysis. In this study, long-term outcomes were certainly affected by other factors, namely: Hunt–Hess grade, Fisher scale score, and the time of treatment.

The immanent limitation of the studies examining the morphometric parameters of RIA comes from the fact that DSA shows nothing but contrast medium. The contrast medium in a vessel lumen rarely outlines its interior in a perfect way, whereas, in turn, constructing a 3D model of brain vessels is radiologist-dependent. For example, the morphometry of a ruptured aneurysm may be affected by an error resulting from the presence of a clot in its sac, rendering it impossible to assess its true shape.

As in every numerical research, there are several assumptions and limitations that do not resemble the realistic, physiological behavior of the analyzed case. The main simplifications utilized within this study included rigid walls of arteries and aneurysm, specific non-Newtonian shear-thinning blood model, and homogenous porosity of the aneurysm dome. Unfortunately, the spatial alignment of the coil (and consequently its packing density) is extremely random; hence, it is impossible to predict how it aligns within the aneurysm dome during a surgical procedure. Therefore, the authors assumed that coil packing density is uniform across the entire aneurysm.

Another limitation of our work is its single-center setting. Consequently, a possibility of selection bias and measurement bias cannot be excluded. To conclude, our results are preliminary and require multicenter prospective studies. This study offers a step towards personalizing endovascular therapy and building guidelines for a patient-tailored choice of endovascular procedures.

## 5. Conclusions

We present an easy intraoperatively calculable predictor which has a potential to be used in clinical practice as a means of improving the outcome. According to our results, achieving 1st VPD of more than 10.56% decreases the late recanalization rate. The proposed predictor is not based on a complex model comprising numerous markers, but on a single measurement which may be easily done intraoperatively and directly interpreted. If we achieve high 1st VPD values, this could, in practice, help to decrease the occurrence of the late recanalization and decrease the retreatment rate, and the patient’s exposure to radiation.

## Figures and Tables

**Figure 1 jcm-10-01348-f001:**
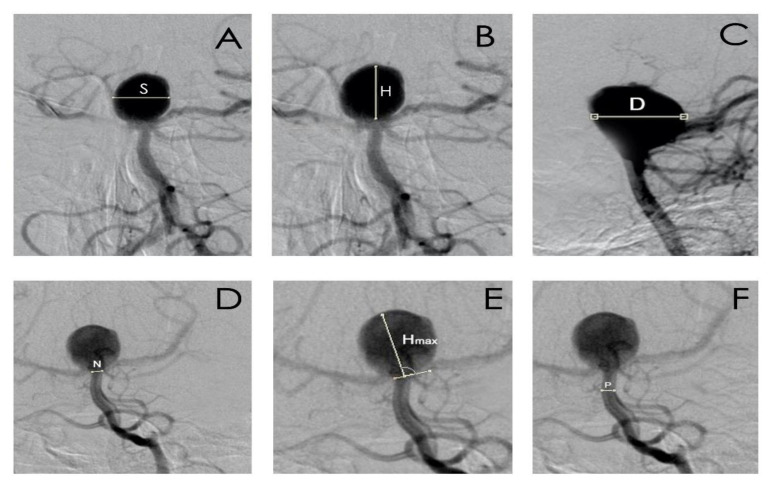
The morphometric parameters: (**A**). S—the aneurysm dome width size, (**B**). H—the aneurysm dome height size, (**C**). D—the aneurysm dome depth size, (**D**). N—aneurysm neck size, (**E**). H_max_—maximal perpendicular height, the largest perpendicular distance from the neck of the aneurysm to the dome of the aneurysm, (**F**). P—parent artery diameter.

**Figure 2 jcm-10-01348-f002:**
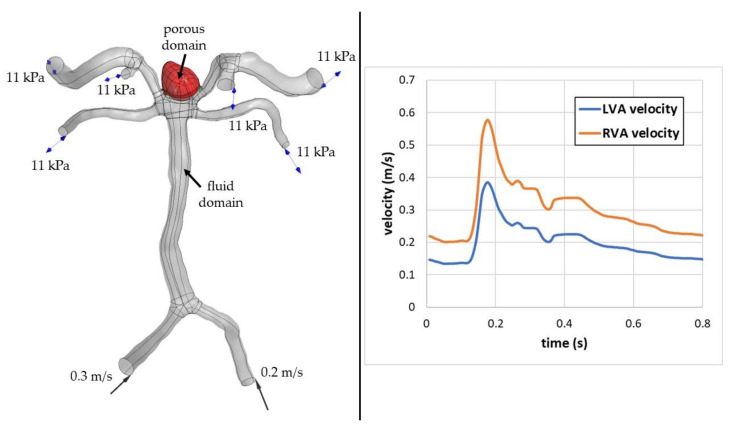
Numerical domain of the analysed case and transient inflow boundary conditions.

**Figure 3 jcm-10-01348-f003:**
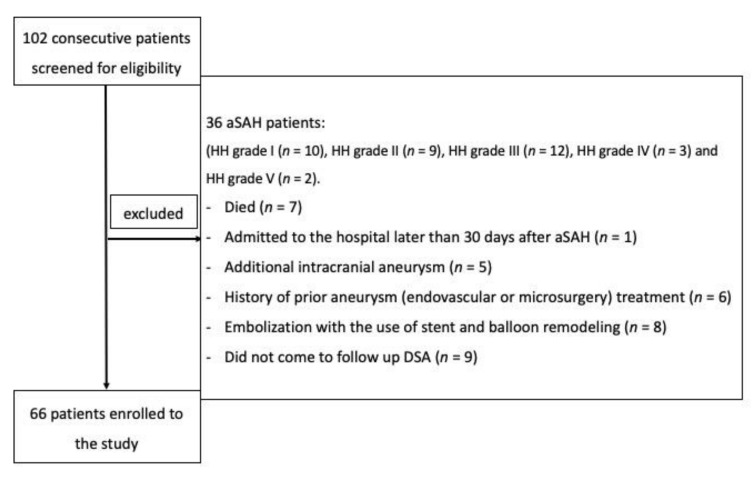
Flowchart presenting the enrollment of the patients.

**Figure 4 jcm-10-01348-f004:**
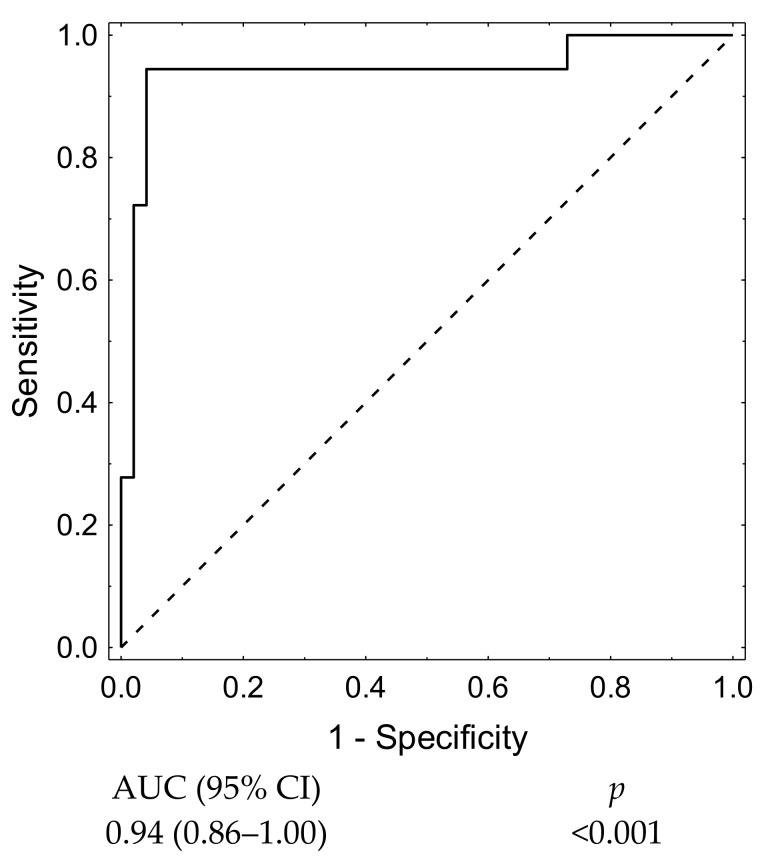
1st VPD receiver operating characteristic (ROC) curve.

**Figure 5 jcm-10-01348-f005:**
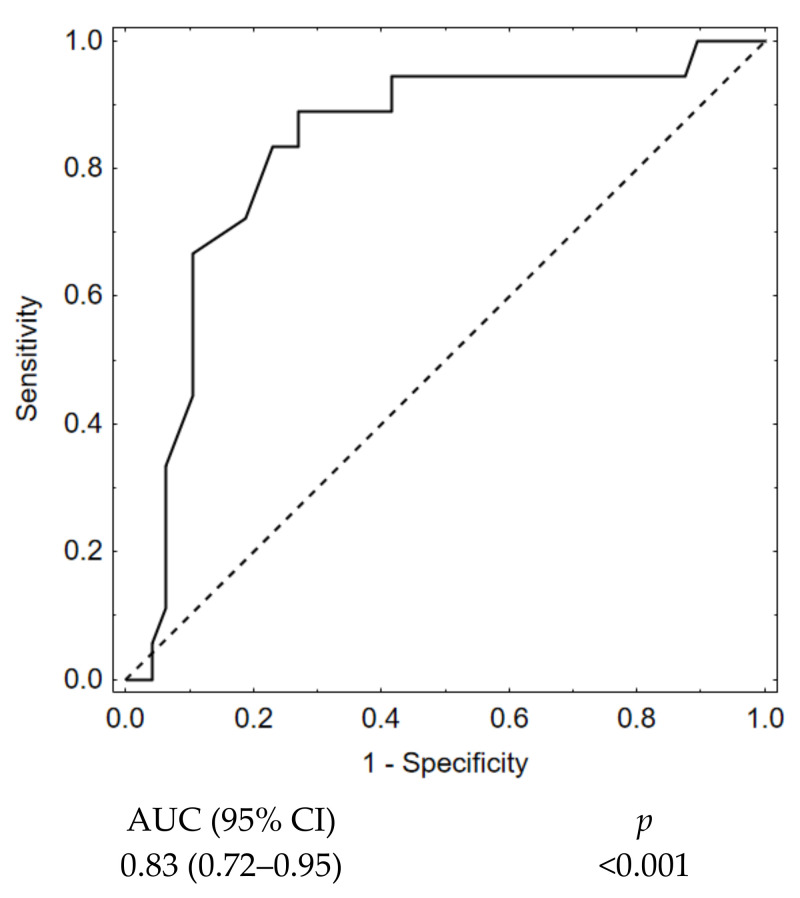
Aneurysm neck size ROC curve.

**Figure 6 jcm-10-01348-f006:**
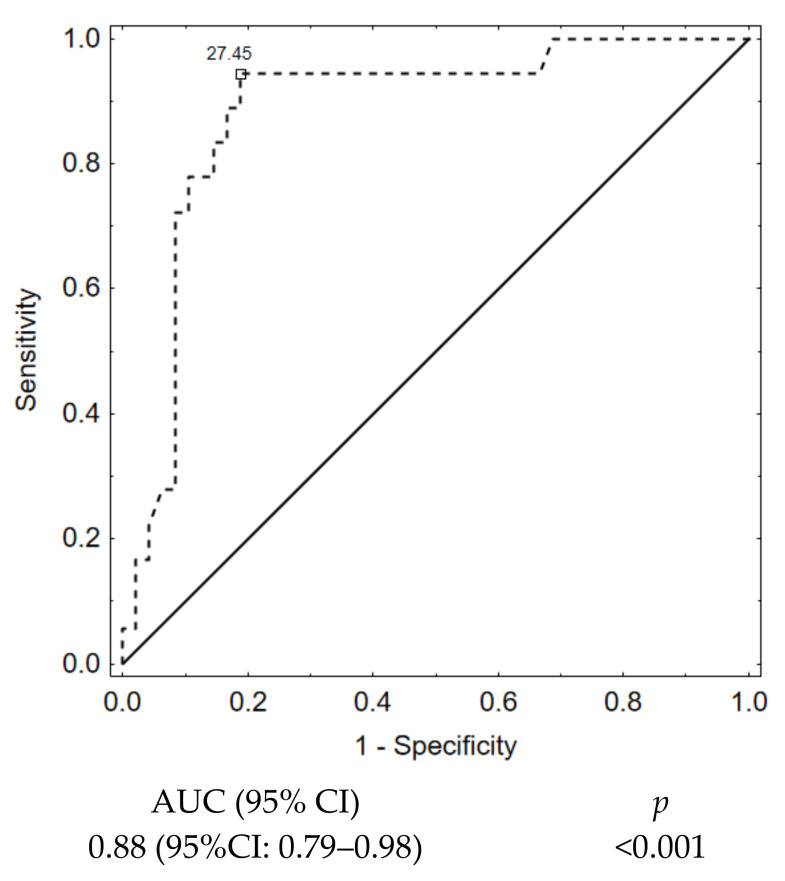
Packing density ROC curve.

**Figure 7 jcm-10-01348-f007:**
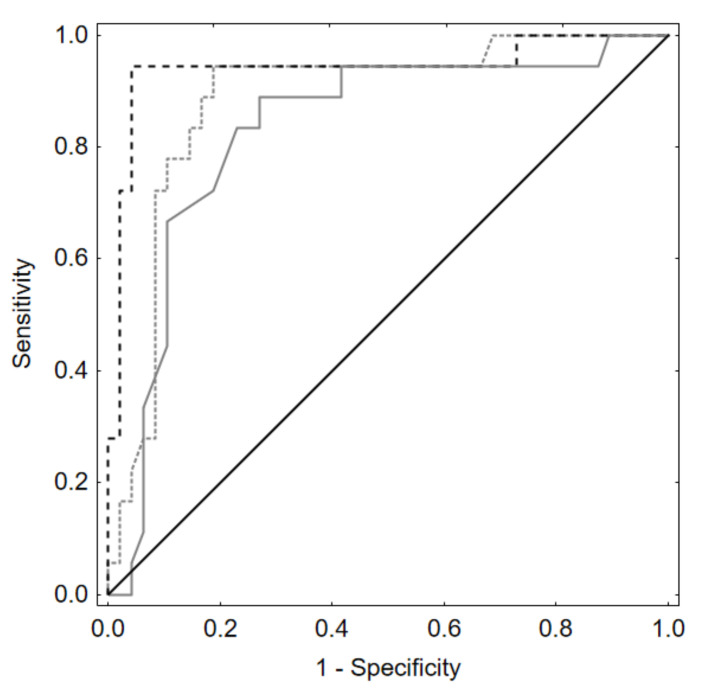
ROC curves based on backward stepwise logistic regression models. Prediction of late recanalization by neck size, packing density, and 1st VPD. 1st VPD—dashed line. Aneurysm neck size alone—solid line. Packing density—line in-between.

**Figure 8 jcm-10-01348-f008:**
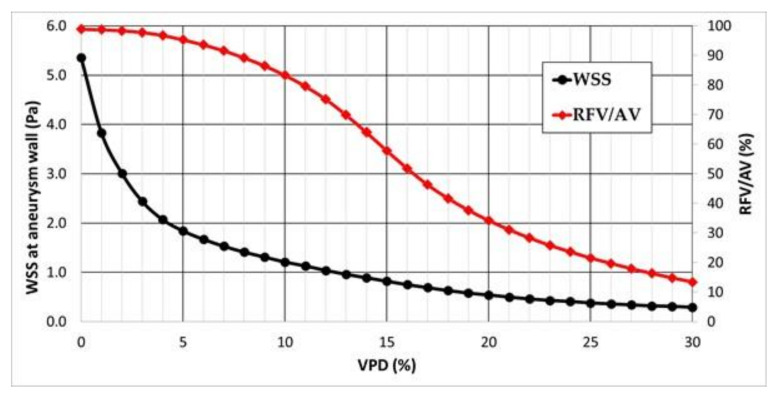
Area-averaged WSS at the aneurysm wall and residual flow volume (RFV)/aneurysm volume (AV) ratio values for each analysed VPD; stationary simulations.

**Figure 9 jcm-10-01348-f009:**
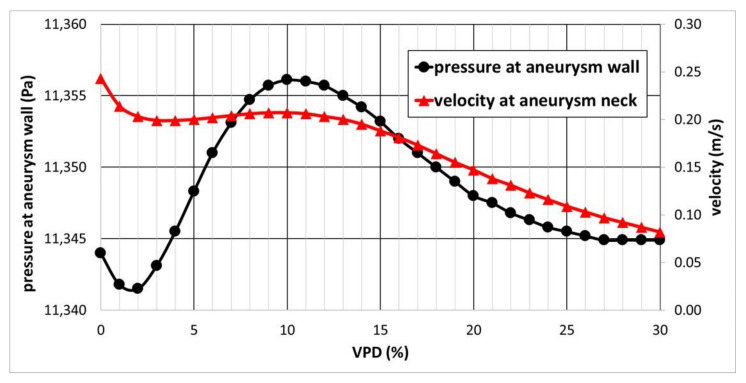
Area-averaged pressure at the aneurysm wall and area-averaged velocity at the aneurysm neck for each analyzed VPD; stationary simulations.

**Figure 10 jcm-10-01348-f010:**
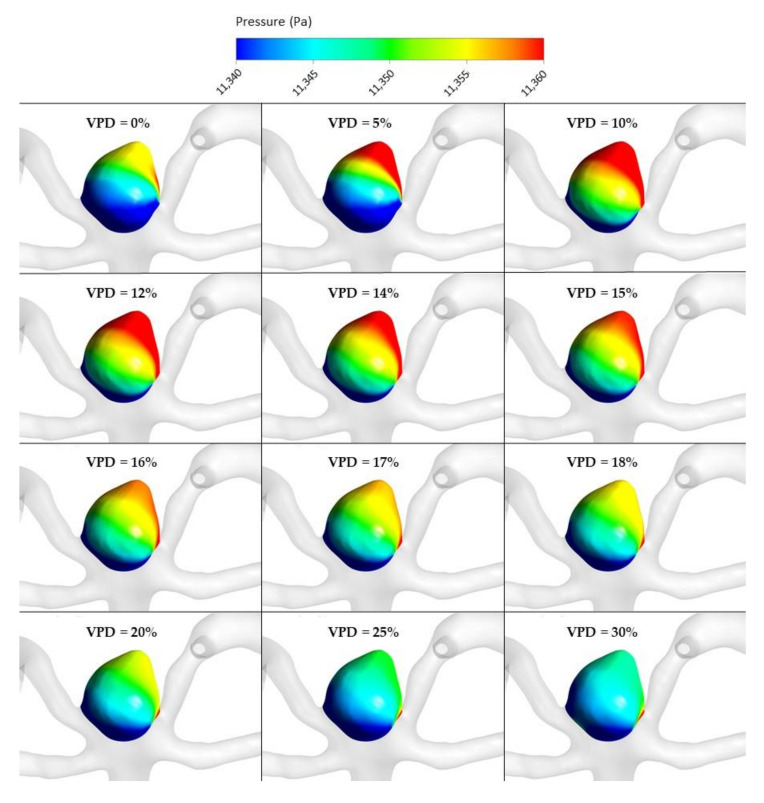
Pressure distribution at aneurysm wall for selected VPD cases; stationary simulations.

**Figure 11 jcm-10-01348-f011:**
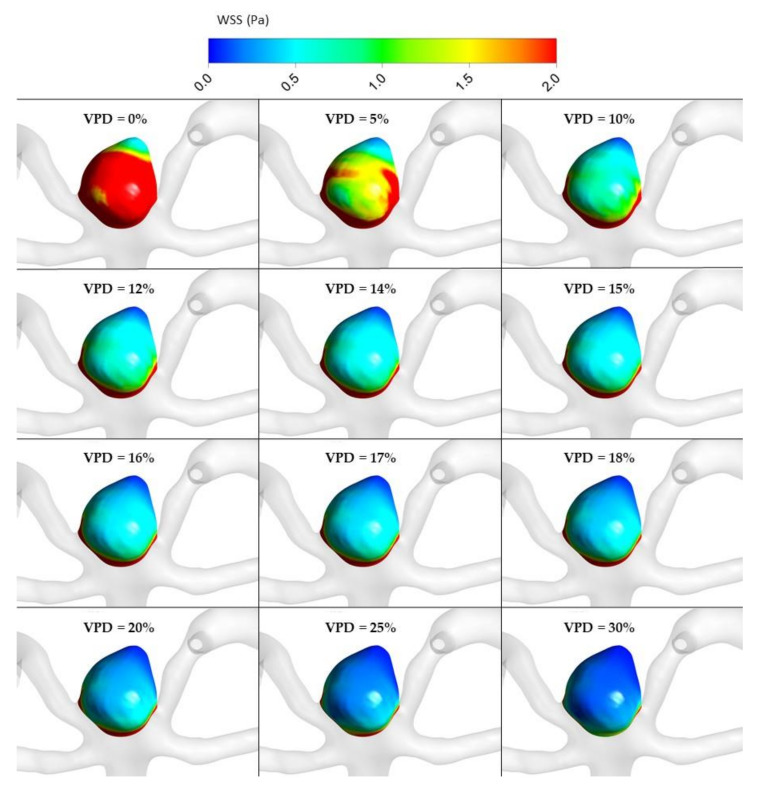
WSS distribution at aneurysm wall for selected VPD cases; stationary simulations.

**Figure 12 jcm-10-01348-f012:**
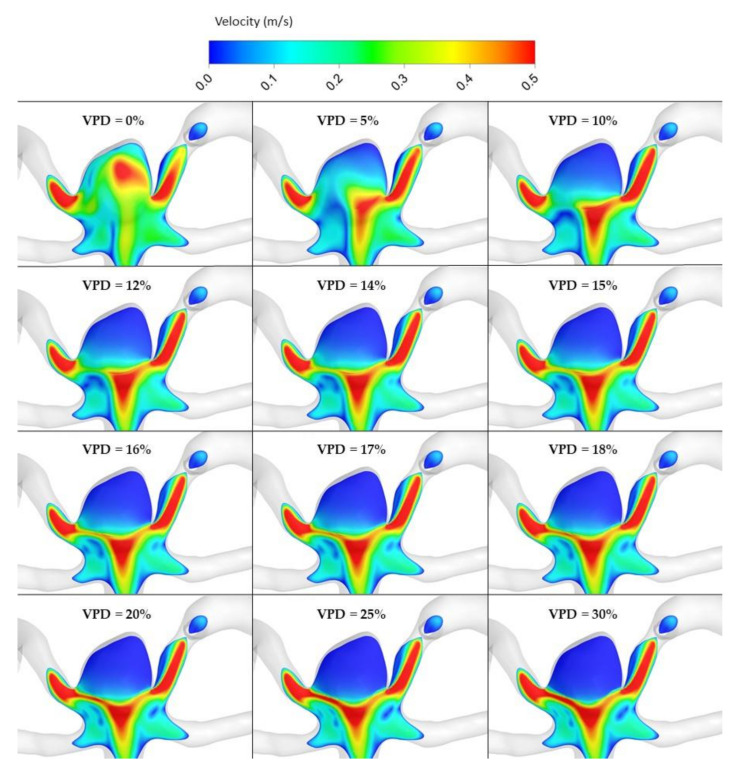
Velocity distribution at aneurysm cross section for selected VPD cases; stationary simulations.

**Figure 13 jcm-10-01348-f013:**
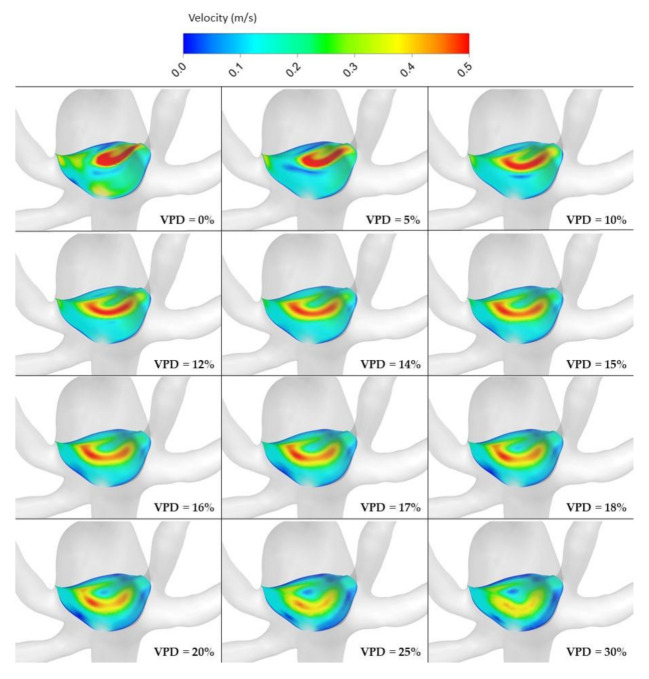
Velocity distribution at aneurysm neck for selected VPD cases; stationary simulations.

**Figure 14 jcm-10-01348-f014:**
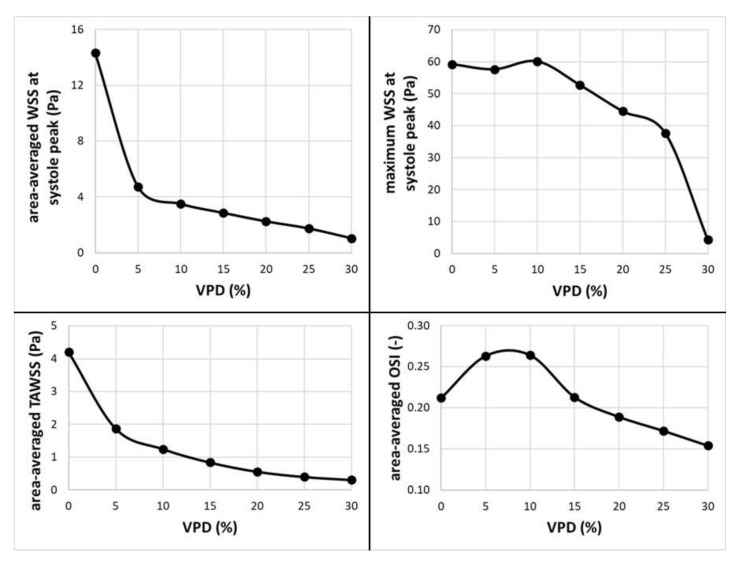
Dependence of selected hemodynamic parameters (at aneurysmal wall) on the 1st coil packing density; transient simulations.

**Figure 15 jcm-10-01348-f015:**
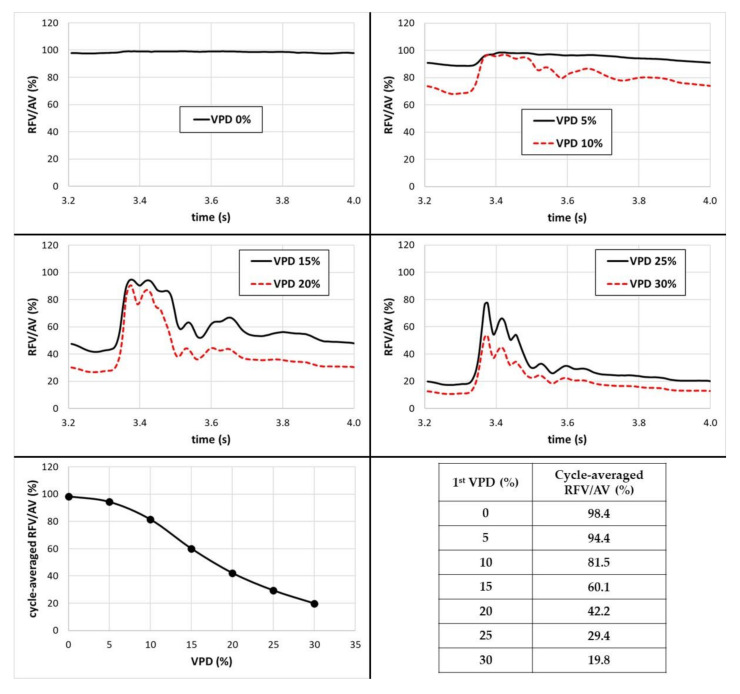
Time-dependent RFV/AV ratio together with cycle-averaged values.

**Figure 16 jcm-10-01348-f016:**
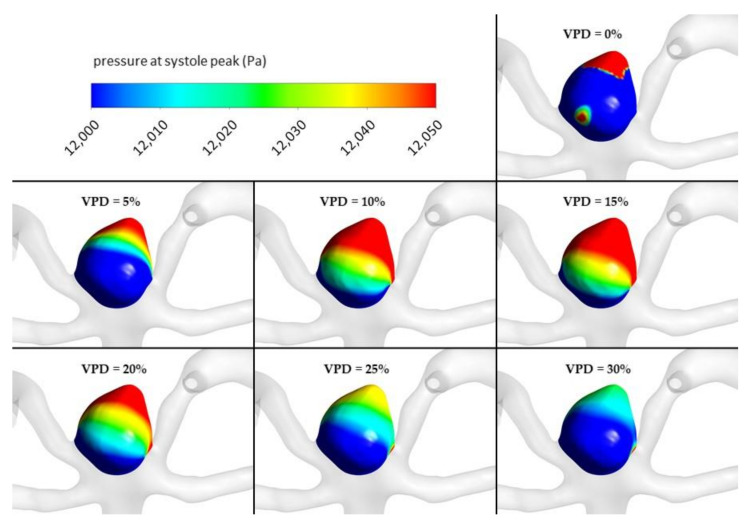
Pressure distribution at aneurysm wall for selected VPD cases; transient simulations—systole peak.

**Figure 17 jcm-10-01348-f017:**
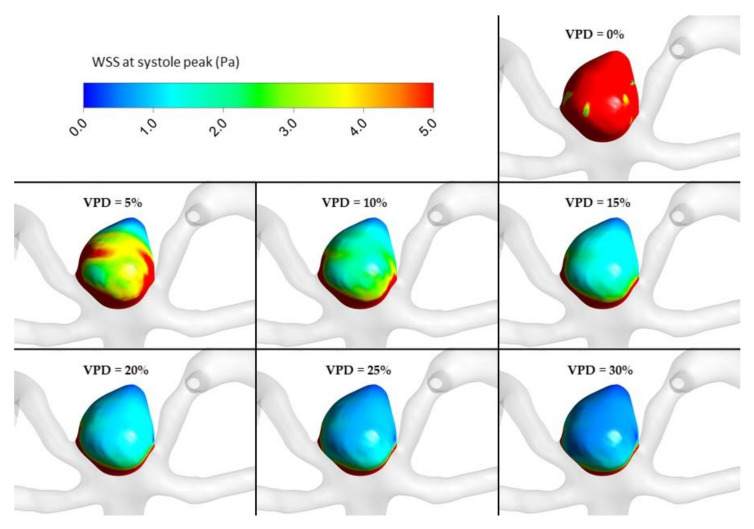
WSS distribution at aneurysm wall for selected VPD cases; transient simulations—systole peak.

**Figure 18 jcm-10-01348-f018:**
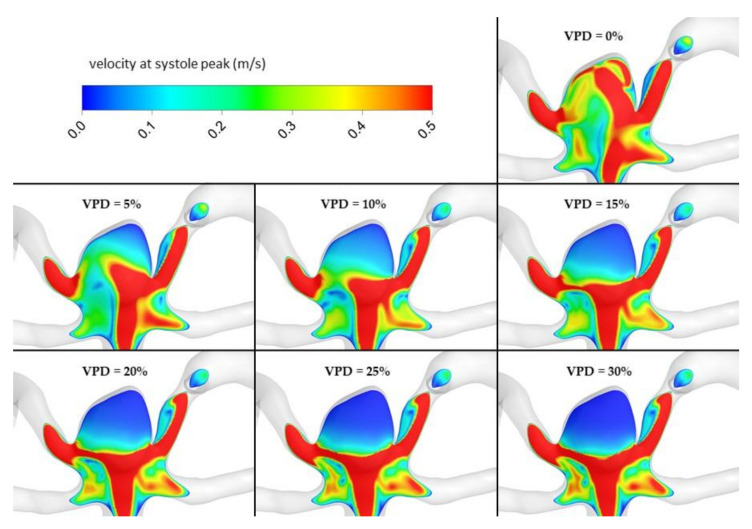
Velocity distribution at aneurysm cross section for selected VPD cases; transient simulations—systole peak.

**Figure 19 jcm-10-01348-f019:**
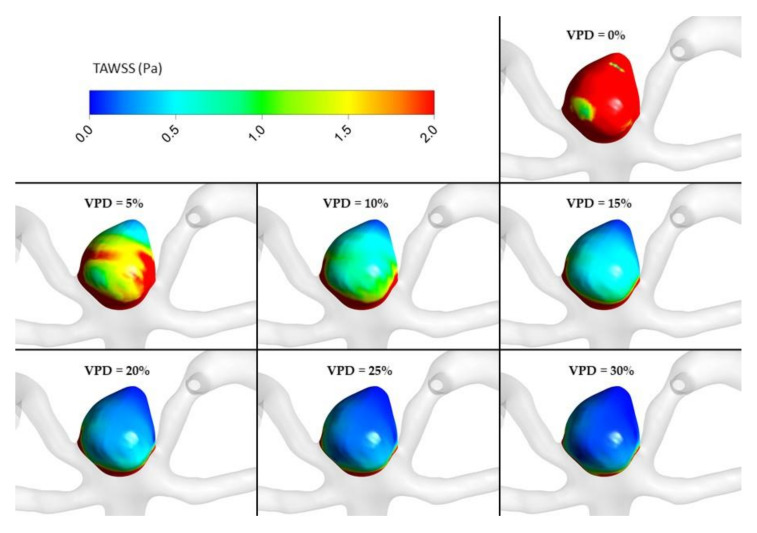
TAWSS distribution at aneurysm wall for selected VPD cases; transient simulations—cycle-averaged values.

**Figure 20 jcm-10-01348-f020:**
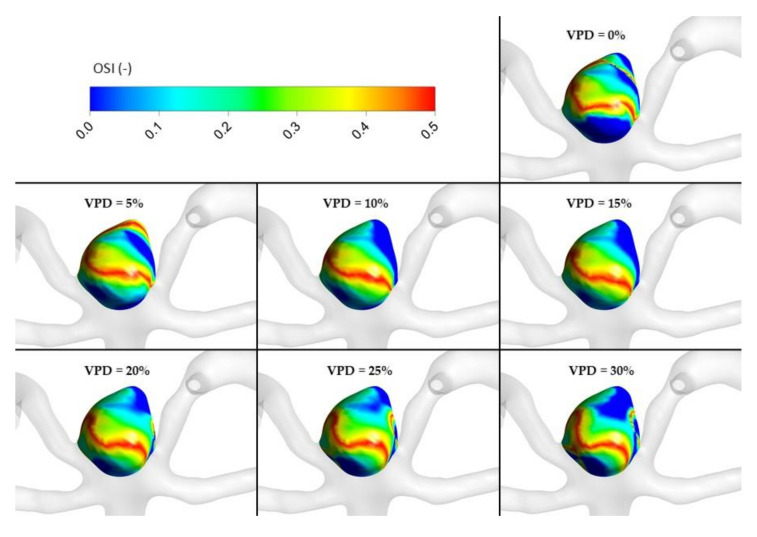
OSI distribution at aneurysm wall for selected VPD cases; transient simulations—cycle-averaged values.

**Table 1 jcm-10-01348-t001:** Inclusion and exclusion criteria.

	Study Group
Inclusion criteria	Patients with SAH from a saccular aneurysm Patients in whom the coil embolization was carried out without the use of stent and balloon remodeling Patients followed for at least a year in whom check-up DSA was done 6 and 12 months after the treatment
Exclusion criteria	Patients with multiple intracranial aneurysms Patients admitted to hospital later than 30 days after aSAH Patients with a history of prior aneurysm (endovascular or microsurgery) treatment

**Table 2 jcm-10-01348-t002:** Mesh independence test for the analyzed geometry.

*N_1_*–14039328 elements *N_2_*–6237876 elements *N_3_*–2436595 elements			*r_21_*–1.310 *r_32_*–1.368		
	**Right posterior cerebral artery; velocity (m/s)**	**Left superior cerebellar artery;** **velocity (m/s)**	**Median cross section of the aneurysm;** **velocity (m/s)**	**Frontal cross section of the aneurysm;** **velocity (m/s)**	**Aneurysm walls;** **WSS (Pa)**
${ø}_{1}$	0.1334	0.1391	0.182	0.221	3.82
${ø}_{2}$	0.1328	0.1388	0.179	0.218	3.83
${ø}_{3}$	0.1340	0.1375	0.169	0.19	3.74
*p*	2.319	4.450	3.575	6.984	4.219
${ø}_{ext}^{32}$	0.132	0.139	0.184	0.222	4.969
$e_{a}^{32}$	0.90%	0.94%	5.59%	12.84%	0.36%
$e_{ext}^{32}$	0.85%	0.31%	2.63%	1.60%	0.13%
${ø}_{ext}^{21}$	0.134	0.139	0.184	0.222	4.983
$e_{a}^{21}$	0.45%	0.22%	1.65%	1.36%	0.10%
$e_{ext}^{21}$	0.51%	0.09%	1.00%	0.24%	0.05%
$GCI_{middle}^{32}$	**1.29%**	**0.50%**	**4.29%**	**2.86%**	**0.21%**
$GCI_{fine}^{21}$	**0.64%**	**0.12%**	**1.26%**	**0.30%**	**0.06%**

**Table 3 jcm-10-01348-t003:** Assessment of 66 patients with ruptured aneurysms.

Type of Scale	Numer of Cases
**Hunt-Hess grade**	***n* = 66**
I	11
II	26
III	24
IV	5
V	0
**Fisher grade ***	***n* = 66**
I	11
II	23
III	15
IV	13
**Glasgow Outcome Scale (GOS) at discharge**	***n* = 66**
V	13
IV	32
III	10
II	7
I	4
**Modified Rankin scale (after 12 moths)**	***n* = 66**
1	22
2	16
3	6
4	9
5	9
6	4
**Comorbidities** (mostly arterial hypertension, diabetes mellitus-DM (type I or II))	47/66

* SAH on CT was confirmed in 62 (93.9%) patients. In 4 patients (6%), CT did not reveal blood, and SAH was confirmed on lumbar puncture.

**Table 4 jcm-10-01348-t004:** Laboratory results, morphometric parameters, packing density, and first coil volume packing density (1st VPD) of intracranial aneurysms.

Value	Mean ± SD without Recanalization	Mean ± SD with Recanalization	*p*
Age (years)	56.75 ± 15.28	56.89 ± 16.12	0.974
depth (mm)	5.46 ± 2.68	6.42 ± 3.29	0.229
height (mm)	6.86 ± 3.22	9.49 ± 5.19	0.016
width (mm)	5.37 ± 2.63	6.63 ± 3.31	0.111
neck size (mm)	3.27 ± 0.83	4.14 ± 0.66	<0.001 *
APTT	30.09 ± 5.51	28.97 ± 2.86	0.473
INR	1.07 ± 0.28	1.08 ± 0.13	0.916
HCT	35.55 ± 11.42	36.49 ± 10.09	0.759
APTT ratio	96.64 ± 14.03	95.19 ± 8.26	0.683
aneurysm volume (mm^3^)	166.48 ± 274.3	349.15 ± 432.39	0.045
packing density (%)	35.0 ± 10.8	21.2 ± 6.6	<0.001 *
1st VPD (%)	18.28 ± 4.16	10.51 ± 2.83	<0.001 *
Parent artery diameter (mm)	3.75 ± 0.92	3.92 ± 1.08	0.525
The largest aneurysm size (mm)	7.04 ± 3.16	9.44 ± 5.17	0.026
SR (maximum aneurysm height divided by the parent artery diameter)	2.02 ± 1.12	2.49 ± 1.27	0.147
index determining the ratio of neck width to diameter of the parent artery	0.92 ± 0.29	1.16 ± 0.46	0.015
Hmax - the largest aneurysm dimension perpendicular to the neck (mm)	6.89 ± 3.41	12.03 ± 5.09	<0.001 *
aspect ratio (AR)—index determining the ratio of the largest dimension of the aneurysm perpendicular to the neck to the width of the aneurysm neck	2.14 ± 0.90	3.03 ± 1.59	0.006

* significant after Bonferroni correction.

**Table 5 jcm-10-01348-t005:** Multivariate analysis.

Variable	Odds Ratio (95% CI)	*p* Value
**Multivariate analysis for late recanalization**		
1st VPD	0.50 (0.35–0.72)	<0.001
**Multivariate analysis for long-term outcome**		
GOS at discharge	0.06 (0.01–0.28)	<0.001

**Table 6 jcm-10-01348-t006:** Numerical results obtained for each 1st VPD value; stationary simulations.

1st VPD (%)	Velocity at the Aneurysm Neck (m/s)	WSS at the Artery Wall (Pa)	WSS at the Aneurysm Wall (Pa)	Pressure at the Aneurysm Wall (Pa)	RFV/AV (%)
**0**	0.243	4.99	5.36	11,344.0	98.9
**1**	0.214	4.98	3.83	11,341.8	98.7
**2**	0.203	4.98	3.01	11,341.5	98.4
**3**	0.199	4.98	2.44	11,343.1	97.8
**4**	0.199	4.98	2.07	11,345.5	96.8
**5**	0.200	4.99	1.84	11,348.3	95.3
**6**	0.202	4.99	1.67	11,351.0	93.6
**7**	0.204	5.00	1.53	11,353.1	91.6
**8**	0.206	5.00	1.41	11,354.7	89.2
**9**	0.207	5.01	1.31	11,355.7	86.5
**10**	0.207	5.01	1.21	11,356.1	83.3
**11**	0.206	5.02	1.13	11,356.0	79.6
**12**	0.203	5.02	1.04	11,355.7	75.2
**13**	0.200	5.02	0.96	11,355.0	69.9
**14**	0.195	5.03	0.89	11,354.2	64.0
**15**	0.188	5.03	0.82	11,353.2	57.8
**16**	0.181	5.03	0.75	11,352.0	51.8
**17**	0.173	5.04	0.69	11,351.0	46.3
**18**	0.164	5.04	0.63	11,350.0	41.7
**19**	0.155	5.05	0.58	11,349.0	37.7
**20**	0.147	5.05	0.54	11,348.0	34.2
**21**	0.138	5.06	0.50	11,347.5	31.1
**22**	0.131	5.06	0.46	11,346.8	28.3
**23**	0.123	5.06	0.43	11,346.3	25.8
**24**	0.116	5.07	0.41	11,345.8	23.6
**25**	0.109	5.07	0.38	11,345.5	21.5
**26**	0.103	5.07	0.36	11,345.2	19.7
**27**	0.097	5.08	0.34	11,344.9	17.9
**28**	0.092	5.08	0.32	11,344.9	16.3
**29**	0.087	5.08	0.31	11,344.9	14.8
**30**	0.082	5.08	0.29	11,344.9	13.4

## Data Availability

Data available in a publicly accessible repository. The data presented in this study are openly available in open research data repository (Zenondo).

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
