# Peer review of "Porous Media Computational Fluid Dynamics and the Role of the First Coil in the Embolization of Ruptured Intracranial Aneurysms"

_jcm, 2021, doi:10.3390/jcm10071348_

Round 1
Reviewer 1 Report
Authors have conducted a thorough, sound and extensive labour of both basic and clinical research. Authors sought to assess risk factors of recanalization in aSAH patients regarding to radiological data. In addition a computational fluid dynamics model was created to illustrate and explain the significance and meaning of their findings. Their findings support the actual body of knowledge and although their results are not quite innovative their methods are solid, and the investigation has been beautifully conducted and reported.
There are only a few issues that should be addressed prior to endorse this article for publications:
- Aneurysms location should be disclosed since, despite their findings, there exists the assumptions that aneurysms in different locations tend to recur more than others and risk of rupture might be higher.
- A short explanation detailing the procedure to build the model, as well as implemented software, kind of source of this software(open or paying), would be desired. A short orientation on how this models can be accessed by the audience would be also appreciated.
In all, this is a high quality article.
Author Response
28th February 2021
Karol Wiśniewski MD PhD
Department of Neurosurgery and Neurooncology
Medical University of Lodz
22 Kopcińskiego Street, 90-153 Łódź, Poland
Reviewer #1 of Journal of Clinical Medicine,
I am pleased to submit the revised version of the manuscript entitled “Porous media computational fluid dynamics and the role of the first coil in the embolization of ruptured intracranial aneurysms”. We highly appreciate your time and we would like to express our gratitude for the valuable criticism and suggestions. We addressed all your comments and performed a thorough language correction. All changes are highlighted in the revised manuscript. We also provide point-by-point responses to your comments which can be found below.
- Aneurysms location should be disclosed since, despite their findings, there exists the assumptions that aneurysms in different locations tend to recur more than others and risk of rupture might be higher.
We do agree that the aneurysm location might play a significant role when it comes to a recanalization process and rupture, hence, this hypothesis should be investigated. We are grateful for pointing out that we did not provide information on the investigated aneurysm locations in the manuscript. Basing on our prepared statistical database, we included this information to the Results section (please see below).
(…) Among 66 ruptured aneurysms, 42 (63.6%) were located in the anterior part of the circle of Willis. There were 11 aneurysms on ACoA, 4 on MCA, 27 on ICA, including 12 PCoA aneurysm (no fetal type PCoA), 8 on ICA bifurcation, 4 on the ophthalmic segment, 3 on the anterior choroid artery.
Twenty four aneurysms (39.4%) were located in the posterior part of the circle of Willis with 18 of them on BA (bifurcation), 1 aneurysms on PICA, 1 on the posterior cerebral artery, 1 on the superior cerebellar artery, and 3 aneurysms located on the vertebral artery.
Basing on our research and statistical database, we did not find any correlation between aneurysm location and recanalization. (…)
Moreover, we modified a description of the patient-specific model used for CFD analyses to clearly present information concerning the aneurysm location, i.e. at basilar artery. Please see below the modified part of the manuscript:
(…) it was decided to limit the geometry only to the basilar artery aneurysm together with efferent and afferent vessels (vertebral, basilar, posterior cerebral and posterior cerebellar arteries). (…)
- A short explanation detailing the procedure to build the model, as well as implemented software, kind of source of this software (open or paying), would be desired. A short orientation on how this models can be accessed by the audience would be also appreciated.
Initially, we decided not to include this information in the main manuscript body to limit the word count and to keep the manuscript as concise as possible. However, we do think and agree that all information presenting investigated case studies are relevant. Thus, we added a paragraph describing model retrieval methods – please see below. Concerning the software used for the model generation, we used a custom-made software that is currently being developed at Institute of Turbomachinery at Lodz University of Technology, Poland. Raw data and models will be available in a publicly accessible repository - Zenondo.
(…) The initial patient-specific geometry was prepared in a custom-created software (Anatomical Model Reconstructor, AMR) basing on biomedical images acquired during standard angio-CT examination. This software is continually being developed in Institute of Turbomachinery at Lodz University of Technology (Poland), thus currently it is only an in-house program. (…)
(…) The digital patient-specific surface model was obtained from a 3D binary mask that was created during an automatic image segmentation technique known as a region growing (or seed growing). Voxels representing blood domain, which intensity matched predefined thresholding criteria, were iteratively added to an initial region selected by the user. To enhance the output of such an automatic image segmentation method, a manual 3D mask processing was required. Fortunately, AMR software offers an intuitive approach towards manual manipulation of the binary mask – the user can add or remove a region selected with either lasso or a brush. Thus, if any unnecessary anatomical structure (such as bone fragment or another blood vessel that was too close to the investigated artery) was added to the mask during seed growing method, the authors could easily select and remove it. Afterwards, a surface model could be generated from the spatial mask thanks to marching cubes algorithm. Due to the fact that this surface was voxelated one (characterized by a stair-like structure), it was subjected to an automatic smoothing algorithm, available in AMR software as well. It strives for smoothing the surface wall while minimizing topology changes of the entire structure. Finally, distal parts of the efferent and proximal parts of the afferent vessels were clipped to obtain an opened model whose inlet and outlet cross sections were as perpendicular to the flow channel as possible. This was performed in AMR software as well. (…)
Moreover, basing on suggestions of Reviewer #2 we provided numerous additional information in the manuscript as well as we conducted, analyzed and described transient simulations mimicking pulsatile heart nature. Thus, we could gather more realistic and physiological data for our analysis.
To sum up, we hope that our replies are satisfactory and that the revised version of the manuscript is suitable for publication in Journal of Clinical Medicine.
We look forward to your response.
Yours sincerely,
Karol Wiśniewski

Reviewer 2 Report
Title: Porous media computational fluid dynamics and the role of the first coil in the embolization of ruptured intracranial aneurysms
Summary: This study provides a very valuable clinical cohort of ruptured aneurysms treated with coils, and their recurrence outcome at follow-up. Their morphological results show that aneurysms that recur have large necks, have low packing density and lower 1st VPD. These are great results. Furthermore, the authors perform CFD analysis on one aneurysm to simulate the blood-flow and found that after a limit, increase in VPD does not affect the hemodynamic reduction. This is a good study, but lacks further analysis that can do justice to the clinical data-set. Following are my comments:
Comments:
- This study seems like two separate analyses combined into one. One is the clinical study with association of recanalization of ruptured aneurysms with aneurysmal geometrical parameters, which has good results that agree with the literature. Then the authors decided to perform a CFD based sensitivity study on one case (?), which does not really add to the existing literature. I would like to ask the authors to kindly explain this.
- I would suggest that the authors model the exact clinical coiling procedure for all the aneurysm cases and then identify hemodynamics that correlates with recurrence of coiled aneurysms. This analysis will be really useful to test whether image-based CFD analyses could aid in clinical recurrence predictions, thus make this study much stronger.
- The motivation of the study is weak. Authors talk about assessing the effect of the 1st coil volume for ruptured aneurysms, without any substantial background of the clinical significance of the 1st coil volume. This should be addressed in the Introduction.
- To add to my 1st point, I do not see the point of using CFD analyses in this manuscript. There are a few drawbacks of CFD assessment:
- CFD simulations are steady-state: this is a very big assumption.
- WSS and pressure are not the only hemodynamic variables that correlate with recanalization of coiled aneurysms in the literature. The authors should also look at variables like velocity at the aneurysm neck and the aneurysm dome.
- Unsteady simulations could also provide information about OSI, which has been linked to associate with inflammation at the aneurysmal wall. Such parameters should also be considered.
- RFV seems like an arbitrary parameter. Do the authors have references and rationale behind calculating this parameter.
- The hemodynamic results are intuitive and do not add anything clinically important in my opinion. It is well-known that low packing density is associated with recurrence of coiled aneurysms.
- The authors claim that this is the first study that shows 1st VPD identifies residual blood-flow and could predict recanalization after coiling- this statement is debatable. The CFD results have not been correlated with the clinical outcome of the patients, so it cannot be claimed that recanalization prediction is possible using residual blood-flow calculations
- Please provide details of your statistical results in the Results section (Tables 4 and 5); just providing a Table with variables is not sufficient, need some interpretation in the section.
Author Response
28th February 2021
Karol Wiśniewski MD PhD
Department of Neurosurgery and Neurooncology
Medical University of Lodz
22 Kopcińskiego Street, 90-153 Łódź, Poland
Reviewer #1 of Journal of Clinical Medicine,
I am pleased to submit the revised version of the manuscript entitled “Porous media computational fluid dynamics and the role of the first coil in the embolization of ruptured intracranial aneurysms”. We highly appreciate your time and we would like to express our gratitude for the valuable criticism and suggestions. We addressed all your comments and performed a thorough language correction. All changes are highlighted in the revised manuscript. We also provide the point-by-point responses to your comments.
- This study seems like two separate analyses combined into one. One is the clinical study with association of recanalization of ruptured aneurysms with aneurysmal geometrical parameters, which has good results that agree with the literature. Then the authors decided to perform a CFD based sensitivity study on one case (?), which does not really add to the existing literature. I would like to ask the authors to kindly explain this. I would suggest that the authors model the exact clinical coiling procedure for all the aneurysm cases and then identify hemodynamics that correlates with recurrence of coiled aneurysms. This analysis will be really useful to test whether image-based CFD analyses could aid in clinical recurrence predictions, thus make this study much stronger.
We decided to perform the CFD analysis to provide mechanistic insights into our statistically observed clinical results. In other words, CFD was just a tool to visualize our statistical results and to validate whether statistically estimated data could be predicted by CFD. Our goal was to check the hemodynamic parameters in porous media CFD model with wide range of porosities that resembled 1-30% of VPD, with 1% increment for stationary simulations. In total, we performed 31 different CFD analyses (based on a single patient-related aneurysm model) to check how the hemodynamic parameters changed. Following one of your suggestions, we conducted an additional sequence of transient simulations (mimicking pulsatile nature of the heart, where 5 full cardiac cycles were simulated). However, due to limited time given us to provide responses to the Journal, we had to limit time-dependent simulations to just selected 1st VPD values, i.e. 0%, 5%, 10%, 15%, 20%, 25% and 30%. Such increment value allowed us to investigate desired range of VPD in relatively short time period.
CFD provides a vast amount of data with high spatial and time resolution that cannot be obtained during the clinical trials, hence, verifying its validity on ‘artificial cases’ might indicate that CFD could be used preoperatively in patient-specific cases.
In our CFD analysis we have noted that after exceeding a certain threshold value, further increase in VPD does not have a significant influence on the obtained WSS results and RFV/AV ratio for the marginal values of VPD is relatively constant, whereas the highest changes occur in the range 10-20% of VPD. Within these threshold values one can observe the highest inflow reduction to the aneurysmal dome. Thus, it suggests that 1st VPD value should be in 10-20% range which is in line with our clinical data.
In CFD analysis we want to emphasize that after exceeding 10% of 1st VPD, hemodynamic parameters change crucially and that such a phenomenon is in agreement with our clinical data (>10.56%). We are aware that one cannot draw clinical conclusions from CFD analysis at this stage of the research. Thus, in the future we plan to identify hemodynamics that correlates with recurrence of coiled aneurysms.
As far as numerical analyses for all aneurysms are taken into account, we are convinced that this would be the best solution to validate statistical results. Such a broad range of CFD investigations could even serve as an additional statistical database. However, conducting well-defined numerical simulations requires high amount of time and workload, for instance: model generation and its processing; generation of high-quality mesh that meets all the quality criteria (such as positive mesh independence test); simulation processing and calculation stage (e.g. each transient numerical simulation performed within this research was fully calculated after circa 8 hours); results post processing; preparation of clear illustrations; export of the most relevant hemodynamic parameters in the numerical form and their manipulation to present them in tabularized form. It is worth mentioning that the next obstacle is related to the limited data storage - results of such simulations (with high-quality meshes) occupy significant amount of space in an available computer memory. For instance, transient results only for a last cardiac cycle (obtained during this research) calculated for 7 different VPD values, occupy nearly 250 GB of a disk space. Thus, numerical simulations performed only for 4 similar geometries could result in over 1.0 TB of data to be stored.
- The motivation of the study is weak. Authors talk about assessing the effect of the 1stcoil volume for ruptured aneurysms, without any substantial background of the clinical significance of the 1st coil volume. This should be addressed in the Introduction.
We are grateful for this remark. We added the appropriate paragraph to the introduction section. Please see below.
(…) Many studies concluded that it was necessary to increase the PD by at least 20–24% to avoid aneurysmal recanalization (11,12,22), however, the final decision on coil characteristics and number of coils used during embolization, always depends on the neuroradiologist. On the other hand, some authors claimed that the first coil is crucial for obtaining a high PD and improving outcomes (23,24). It is still unknown how the first coil changes intraaneurysmal hemodynamic parameters and how it influences recanalization. The mechanistic background of the first coil volume packing density (1st VPD) was not well investigated.
In this study, we looked at the role of 1st VPD in the embolization of ruptured intracranial aneurysms (RIAs) in order to identify a strong predictor for recanalization after coil embolization of ruptured aneurysms and to find the basic mechanistic/physical background of our clinical observations. This goal was achieved by means of a statistical analysis followed by a computational fluid dynamics (CFD) with porous media modelling approach, during both stationary and transient simulations (mimicking pulsatile blood flow). (…)
- 23. Mascitelli JR, Patel AB, Polykarpou MF, Patel AA, Moyle H. Analysis of early angiographic outcome using unique large diam- eter coils in comparison with standard coils in the embolization of cerebral aneurysms: a retrospective review. J Neurointerv Surg. 2015;7:126–30
- Khatri R, Chaudhry SA, Rodriguez GJ, Suri MF, Cordina SM, Qureshi AI. Frequency and factors associated with unsuccessful lead (first) coil placement in patients undergoing coil embolization of intracranial aneurysms. Neurosurgery. 2013;72:452–8.
- To add to my 1stpoint, I do not see the point of using CFD analyses in this manuscript. There are a few drawbacks of CFD assessment:
- CFD simulations are steady-state: this is a very big assumption.
We do agree that stationary simulations might be a robust approach towards validating the statistical results. We are aware that steady-state analyses might be an oversimplification, however, they still can provide some robust data. Nonetheless, we decided to expand our research by providing transient simulations mimicking pulsatile nature of the blood flow. In each analysis we simulated five full cardiac cycles and we investigated the following 1st VPD values: 0%, 5%, 10%, 15%, 20%, 25% and 30%. Unfortunately, we could not calculate additional transient simulations with smaller increment due to obligatory time limits of our response to the Journal. Nonetheless, such a range of 1st VPD values analyzed during time-dependent simulations allowed us to validate our statistical results. We achieved a conformity of stationary and transient CFD with our statistical results.
- WSS and pressure are not the only hemodynamic variables that correlate with recanalization of coiled aneurysms in the literature. The authors should also look at variables like velocity at the aneurysm neck and the aneurysm dome.
We expanded the results analysis, both qualitative and quantitative, by investigating WSS, pressure, area-averaged velocity at the aneurysm dome cross-section, area-averaged velocity at the aneurysm neck, Residual Flow Volume for both stationary and transient simulations. Additionally, we focused on TAWSS (time-averaged WSS) and OSI parameter calculated for the last cardiac cycle in each unsteady-state simulation. Apart from tabularized and plot forms of the calculated data, we embedded numerous illustrations in the main manuscript that depict distribution of the chosen hemodynamic parameters. Thus, the reader can investigate how the given parameter is distributed on the aneurysm wall and how it changes with an increase of the 1st VPD.
- Unsteady simulations could also provide information about OSI, which has been linked to associate with inflammation at the aneurysmal wall. Such parameters should also be considered.
We do agree that OSI can give an additional insight concerning the hemodynamic state of the aneurysm. Thus, after performing a series of transient simulations, we analyzed OSI parameter changes resulting from an increase of the 1st VPD. Moreover, we analyzed TAWSS parameter as well. Both parameters are presented in three forms: plot, table and graphical representation, i.e. illustrations.
- RFV seems like an arbitrary parameter. Do the authors have references and rationale behind calculating this parameter.
We thank the Reviewer for this remark – we did not provide any referential towards RFV value and its internal cut-off point. RFV is a parameter presented in literature, suggested by Umeda et al. (2017). We would like to apologize for not providing a corresponding reference. A given part of the manuscript was modified and now a reference is provided.
- Umeda Y, Ishida F, Tsuji M, Furukawa K, Shiba M, Yasuda R, et al. (2017) Computational fluid dynamics (CFD) using porous media modeling predicts recurrence after coiling of cerebral aneurysms. PLoS ONE 12(12): e0190222.\
- The hemodynamic results are intuitive and do not add anything clinically important in my opinion. It is well-known that low packing density is associated with recurrence of coiled aneurysms.
At this stage of evidence, we cannot draw any clinical conclusion according to our CFD analysis. It is also known that high PD prevents from recanalization, but it is unknown why the 1st VPD is a better recanalization predictor than PD and how the 1st VPD changes intraaneurysmal hemodynamic parameters, which are the key in our opinion to recanalization occurrence.
Which coil will be implanted is strictly neuroradiologist-related. It could be easily calculated after the first angiography, thus, it is an easily modifiable parameter which makes it worthwhile scientific attention.
- The authors claim that this is the first study that shows 1stVPD identifies residual blood-flow and could predict recanalization after coiling- this statement is debatable. The CFD results have not been correlated with the clinical outcome of the patients, so it cannot be claimed that recanalization prediction is possible using residual blood-flow calculations
We rephrased the whole sentence for better clarity. Please see below.
(…) This is the first study, based on retrospective analysis and porous media CFD, which demonstrates that 1st VPD is a strong and potentially clinically useful recanalization predictor, and it determines the volume of postcoiling intra-aneurysmal residual blood flow. (…)
- Please provide details of your statistical results in the Results section (Tables 4 and 5); just providing a Table with variables is not sufficient, need some interpretation in the section.
To address this issue, we removed table 5 and added the following paragraphs to results section.
- Laboratory results, morphometric parameters, packing density, 1st VPD of intracranial aneurysms
In the univariate analyses comparing patients with and without recanalization we observed significant differences between: aneurysm height (9.49 ± 5.19 mm vs. 6.86 ± 3.22 mm, p=0.016), aneurysm neck size (4.14 ± 0.66 mm vs. 3.27 ± 0.83 mm, p <0.001), aneurysm volume (349.15 ± 432.39 mm3 vs. 166.48 ± 274.3 mm3, p=0.045), packing density (21.2 ± 6.6 % vs. 35.0 ± 10.8 %, p<0.001), 1st VPD (10.51 ± 2.83 % vs. 18.28 ± 4.16 %, p<0.001), the largest aneurysm size (9.44 ± 5.17 mm vs. 7.04 ± 3.16 mm, p=0.026, index determining the ratio of neck width to diameter of the parent artery (1.16 ± 0.46 vs. 0.92 ± 0.29, p=0.015), Hmax (12.03 ± 5.09 mm vs. 6.89 ± 3.41 mm, p<0.001) and aspect ratio (3.03 ± 1.59 vs. 2.14 ± 0.90, p=0.006). After Bonferroni correction for multiple hypotheses testing, the differences for neck size, packing density, 1st VPD and Hmax remained statistically significant. Detailed results of all performed analyses are presented in Table 4.
Table 4. Laboratory results, morphometric parameters, packing density, 1st VPD of intracranial aneurysms
Value |
Mean ± SD without recanalization |
Mean ± SD with recanalization |
p |
Age (years) |
56.75 ± 15.28 |
56.89 ± 16.12 |
0.974 |
depth (mm) |
5.46 ± 2.68 |
6.42 ± 3.29 |
0.229 |
height (mm) |
6.86 ± 3.22 |
9.49 ± 5.19 |
0.016 |
width (mm) |
5.37 ± 2.63 |
6.63 ± 3.31 |
0.111 |
neck size (mm) |
3.27 ± 0.83 |
4.14 ± 0.66 |
<0.001* |
APTT |
30.09 ± 5.51 |
28.97 ± 2.86 |
0.473 |
INR |
1.07 ± 0.28 |
1.08 ± 0.13 |
0.916 |
HCT |
35.55 ± 11.42 |
36.49 ± 10.09 |
0.759 |
APTT ratio |
96.64 ± 14.03 |
95.19 ± 8.26 |
0.683 |
aneurysm volume (mm3) |
166.48 ± 274.3 |
349.15 ± 432.39 |
0.045 |
packing density (%) |
35.0 ± 10.8 |
21.2 ± 6.6 |
<0.001* |
1st VPD (%) |
18.28 ± 4.16 |
10.51 ± 2.83 |
<0.001* |
Parent artery diameter (mm) |
3.75 ± 0.92 |
3.92 ± 1.08 |
0.525 |
The largest aneurysm size (mm) |
7.04 ± 3.16 |
9.44 ± 5.17 |
0.026 |
SR (maximum aneurysm height divided by the parent artery diameter) |
2.02 ± 1.12 |
2.49 ± 1.27 |
0.147 |
index determining the ratio of neck width to diameter of the parent artery |
0.92 ± 0.29 |
1.16 ± 0.46 |
0.015 |
Hmax - the largest aneurysm dimension perpendicular to the neck (mm) |
6.89 ± 3.41 |
12.03 ± 5.09 |
<0.001* |
aspect ratio (AR) - index determining the ratio of the largest dimension of the aneurysm perpendicular to the neck to the width of the aneurysm neck |
2.14 ± 0.90 |
3.03 ± 1.59 |
0.006 |
* significant after Bonferroni correction
- Complete aneurysm filling during the first embolization
Complete aneurysm filling on the first embolization according to modified Raymond-Roy scale took place in 52 cases (class I – 78,8%). Incomplete aneurysm filling was noted in 14 cases; class II – 11 cases (16 ,7%), class IIIa – 1 case (1,5%) and class IIIb – 2 cases (3%).
To sum up, we hope that the extended numerical simulations including the transient approach and our replies are satisfactory and that the revised version of the manuscript will be appreciated as suitable for publication in Journal of Clinical Medicine.
We look forward to your response.
Yours sincerely,
Karol Wiśniewski
